# Energy Allocation of the Wolf Spider *Pardosa pseudoannulata* under Dietary Restriction

**DOI:** 10.3390/insects14070579

**Published:** 2023-06-25

**Authors:** Yang Zhu, Li Song, Limi Chen, Yueli Yun, Wang Zhang, Yao Zhao, Yu Peng

**Affiliations:** 1Hubei Key Laboratory of Regional Development and Environmental Response, Faculty of Resources and Environmental Science, Hubei University, Wuhan 430062, China; zhu@hubu.edu.cn (Y.Z.); 202021107011078@stu.hubu.edu.cn (L.S.); 2State Key Laboratory of Biocatalysis and Enzyme Engineering, School of Life Sciences, Hubei University, Wuhan 430062, China; 201711110710954@stu.hubu.edu.cn (L.C.); yueliyun@hubu.edu.cn (Y.Y.); zhangxiaowang@stu.hubu.edu.cn (W.Z.)

**Keywords:** dietary restriction, spider, energy allocation, nutrient, predation

## Abstract

**Simple Summary:**

The phenomenon of food shortage is widespread in spider populations, but the response of *Pardosa pseudoannulata* to dietary restriction remains unclear. The study aimed to determine (1) whether energy allocation occurs in *P. pseudoannulata* under dietary restriction by analysing its development and longevity; (2) if there is a trade-off between development, longevity, and nutrient content indicating a trend to energy allocation; (3) how different levels and periods of dietary restriction affect the predatory ability of *P. pseudoannulata*. The results showed that the continuous dietary restriction has negative impacts on the growth of *P. pseudoannulata*, and positive impacts on its longevity. When food was severely restricted in the juvenile stage, nutrient contents of the adult spider could return to the same level as the control group, but the predatory ability decreased. These findings suggest a trade-off between growth, longevity, and predatory ability of the *P. pseudoannulata* under dietary restriction conditions.

**Abstract:**

The phenomenon of food shortage is widespread in spider populations, which has a great impact on their growth, development, and survival. *Pardosa pseudoannulata* is a dominant spider species in rice fields and has an important controlling effect on rice pests. In this study, three feeding levels were tested at the juvenile stage (H, high feeding; M, medium dietary restriction; L, severe dietary restriction) and two at the adult stage (H and L). A total of six feeding levels were tested to explore the effects of dietary restriction on the development, longevity, nutrient content, and predation by *P*. *pseudoannulata* [HH (control group), HL, MH, ML, LH, LL]. The results showed that continuous dietary restriction (ML and LL groups) had negative impacts on the growth of *P*. *pseudoannulata* and positive impacts on longevity. Spiderlings suffered from dietary restrictions during the juvenile period, and when the restrictions were removed upon reaching adulthood (MH and LH groups), their lifespan started decreasing whilst their weight began returning to normal. This suggested that there might be a trade-off between the growth and longevity of the spider under dietary restrictions. The study also found that when food was severely restricted in the juvenile stage (LH and LL groups), the nutrient contents of the adult spider could return to the same level as the control group, but the predatory ability decreased. When food was moderately restricted in the juvenile stage (MH and ML groups), the predatory ability of the adult spiders improved, while nutrients of the adult spiders declined. Our results will provide an empirical basis for the protection and effective use of dominant spider species in agricultural fields.

## 1. Introduction

Food resources are indispensable in most aspects of animal life. Food types and amounts play important roles in growth, development, and survival. However, in the life of animals, the availability of food changes greatly, and the shortage of food resources is widespread. Food restriction is broadly defined as the amount of nutrition or food intake below the normal requirement but above the level of hunger, which is approximately a 20–40% reduction in food intake [1]. There are currently three descriptive categories of food restriction: dietary restriction (DR), calorie restriction (CR), and protein and amino acid restriction [2]. Dietary restriction is a broader concept and includes reducing the amount of food intake while maintaining the original nutrient contents of the food and restricting the intake of specific types of nutrients such as fats or carbohydrates [3]. Dietary restriction is a traditional method used to study the effects of food restriction on survival. Calorie restriction refers to the use of restricting calories for weight reduction, DNA repair, ribosomal and telomere DNA duplication control, mitochondrial regulation, antioxidant activation, and autophagy protection, promoting the health of the organism and increasing the longevity of the organism [4,5]. Protein restriction refers to reducing the intake of a specific protein in food to achieve the effect of food restriction [6,7].

The effect of dietary restriction on animals was first discovered in rodents [8] and has already been proven in yeast, nematodes, flies, mice, and other animals [9,10,11,12,13,14]. In the case of food shortages, animals could deal with food restrictions through energy allocation. Energy allocation is a phenomenon where an individual makes decisions in specific situations. When an individual has limited dietary resources, it chooses to spend its limited energy into growth or longevity in order to ensure the survival and reproduction of the species. Prolonging longevity is a kind of energy allocation at the expense of development and growth. This phenomenon of prolonging longevity has been verified in many species [15,16]. Male and female spiders were also able to induce longevity under dietary restrictions [17,18]. However, the allocation between longevity and body size is not absolutely beneficial, which may increase the risk of death and compromise reproduction. The reproductive effect on guppies (*Poecilia reticulata*) was tested by manipulating food, and researchers found that dietary restriction had no effect on the growth rate or spawning rate, but the number of offspring was significantly reduced [19].

Dietary restriction causes low metabolism and slower digestion, resulting in disrupted growth of the animal [20]. In mammals, in the first stage after severe dietary restriction, the blood sugar, glycogen stores, and fat of the body are used to make up for the lack of nutrients. In the second stage, fat and protein are mainly consumed. When fat consumption reaches a certain critical threshold, the third stage begins, and muscles begin to degenerate [21]. However, studies have shown that different nutritional interventions can have different health benefits for animals. When the diet is lightly restricted by reducing the daily food intake by 20–40%, it has a positive impact on the lifespan of the organism [22]. Therefore, the impact of dietary restriction on organisms depends on not only whether food is restricted or not but also on some factors such as the degree of dietary restriction. The effects of different hunger levels on the lacewing *Chrysoperla carnea* (Stephens) (Neuroptera: Chrysopidae) were different, a higher hunger level led to a longer time to process food [23]. However, the effect of dietary restriction on spiders has rarely been reported [16,17].

As an important living condition for animals, foraging can be affected by many factors. Foraging at elevated rates to provide for offspring is considered an energy-intensive activity, and it is believed to incur a physiological cost for the high workload involved [24]. As the other two important factors for animals, growth, and reproduction can also be affected by many factors, however, when animals suffered a food shortage, different strategies such as energy allocation at the expense of growth and reproduction were used for the effects of dietary restriction on the predation [25]. This is the manifestation of trade-offs between vital life functions under conditions of food shortage. To meet the ever-changing physiological needs of growth and reproduction, animals will adjust their resource acquisition throughout their lives [26]. The behaviour of an animal after food restriction is determined by the trade-off between the value of the food (which increases with the degree of restriction) and other important needs (such as reproduction and avoiding becoming prey) [27]. Yip and Lubin (2016) investigated whether spiders could compensate for diet restriction by altering their foraging behaviour using the orb-weaving spider *Cyrtophora citricola*. They found that spiders were unable to compensate for their restricted diet by altering their foraging behaviour. To date, few studies have explored the effects of dietary restriction on the interactions between growth and foraging under dietary restriction [28].

Since the mid-1950s, various studies about rice field spiders have been carried out, and the role of rice field spiders as an important predatory natural enemy of farmland pests has been recognised by plant protection experts and entomologists [29,30,31]. *Pardosa pseudoannulata* is a kind of wolf spider that is beneficial to rice fields because it is the natural enemy of rice pests [32,33]. To explore the response of *P. pseudoannulata* suffering from dietary restriction, this study conducted a 3 × 2 factorial experiments based on the food level provided at the juvenile and adult stages, and then explored the growth, nutrition, and predatory behaviour of *P*. *pseudoannulata* under these different levels of dietary restriction. The study aimed to determine if: (1) energy allocation occurs in *P*. *pseudoannulata* under dietary restriction by analysing the development and longevity; (2) there is a trade-off between development, longevity, and nutrient content that would indicate a trend to energy allocation; (3) different levels and periods of dietary restriction affect the predatory ability of *P*. *pseudoannulata*. Research on the energy allocation of *P*. *pseudoannulata* can provide an empirical framework to help reduce the workload and cost of indoor spider breeding and has guiding significance for indoor large-scale spider breeding to undertake various ecological and spider investigations. The research on the predation ability of *P*. *pseudoannulata* in this study also has potential practical significance for the rational use of predatory natural enemies for biological control.

## 2. Materials and Methods

### 2.1. Animal Maintenance

Sub-adults of *P*. *pseudoannulata* were collected from rice fields at Huazhong Agricultural University in Wuhan City, Hubei Province in April 2018. This spider can be recognized in the field by the two dark bands on the carapace and ring-like markings on the legs [34,35]. Each spider was maintained in a glass tube (20 mm diameter, 90 mm high) with a moist sponge at the bottom and fed with *Drosophila melanogaster*. Water was added to moisten the sponge twice a week. An absorbent cotton ball was used as a lid. The sub-adults of *P*. *pseudoannulata* were raised to maturity and then mating experiments were conducted. After the female spiders laid eggs, the hatched F1 generation was used for further experiments. The feeding conditions were set at 25 ± 1 °C and 65 ± 5% relative humidity under a 14:10 h (L:D) photoperiod.

### 2.2. Dietary Restriction Setting

The experiment adopted a 3 × 2 full factorial design to control the food of spiderlings and adults. After 5 egg sacs hatched, 45 spiderlings were randomly selected from each egg sac and were randomly divided into three types: high feeding (H; normal feeding); medium dietary restriction (M; mild restriction); severe dietary restriction (L; severe restriction). Each spider was maintained in a glass tube (20 mm diameter, 60 mm high). Spiderlings of the 2nd and 3rd instar were fed with frozen *D*. *melanogaster* and *Tendipes* sp. (Diptera: Chironomidae). After the 4th instar stage, the spiderlings were fed with *D*. *melanogaster* and *Tendipes* sp. The spiderlings in groups H, M, and L were fed 3, 2, or 1 times a week, respectively. The spiders in each group were fed according to the feeding quantity in Appendix A. For each increased instar, the amount of prey increased 1.5 times. As spiders mature, their appearance changes significantly. The palpal organ of male spiders will turn black and the epigynum of female spiders will protrude and turn black. After maturation, the adults were immediately divided into two feeding groups, H and L (H, high feeding; L, severe dietary restriction). Only *D*. *melanogaster* was provided as prey in these two groups. The H and L groups were fed 3 or 1 times a week, respectively. Based on the preliminary experiment of *P*. *pseudoannulata*, the number of *D*. *melanogaster* in the H group was 20, which was twice that in the L group. All the spiders were fed until death. According to the different dietary restriction levels of the juveniles (H, M, and L) and adults (H, L), a total of 6 treatment combinations were set up (HH, HL, MH, ML, LH, and LL). The number of mature spiders of at F1 generation in each treatment group of at least 30.

### 2.3. Experiment 1: Effects of Dietary Restriction on the Growth of P. pseudoannulata

#### 2.3.1. Determination of Developmental Period

After the spiderlings hatched from the egg sacs, the development of the spiderlings was observed each day. The moulting and survival of the spiderlings was recorded every day, and the mature age of the adults was recorded after maturity. The number of adults in the H, M, and L groups was 74, 75, and 70, respectively. A total of 225 individuals were been selected for the experiment, and 219 individuals finally survived for subsequent data analysis. According to the time needed to reach each age, we estimated the impact of various dietary restrictions on the development and survival of specific spiders (Appendix A). Fortunately, the length of time needed for spider development did not alter survival rates in the reduced diet group (M, L group).

#### 2.3.2. Determination of Carapace Length, Carapace Width, Body Length, and Weight

Three days after adults had emerged, 10 males and 10 females were selected randomly in each group, and the spiders were anaesthetised with CO_2_. The spiders were taken out while others kept the feeding regime. Then, the carapace length (distance from the foremost end to the end of the head carapace), carapace width (distance of the widest part of the head carapace), and body length (distance from the front end of the head carapace to the end of the abdomen) were measured under a stereo microscope (SZX7, Olympus, Tokyo, Japan). Take pictures before measuring. After the same time interval feeding, spiders are weighed with an electronic balance (BT1251, Sartorius Scientific Instruments, Beijing, China). The feeding experiment was reconducted after measurements.

### 2.4. Experiment 2: Effects of Dietary Restriction on Longevity of P. pseudoannulata

Thirty spiderlings were randomly selected from each of the 6 experimental groups [HH (control group), HL, MH, ML, LH, LL], and the developmental period (time required to moult at each age) and longevity were recorded every day to maturity. Adult longevity refers to the time a spider lived from maturity to death. Total longevity refers to the length of time from hatching to death.

### 2.5. Experiment 3: Effects of Dietary Restriction on the Predation Function of P. pseudoannulata

Before the predation experiments, 15 *P*. *pseudoannulata* adults aged 10 days were randomly selected from each of the 6 experimental groups and were starved for 48 h. Five density treatments of fruit flies were set (N = 10, 20, 30, 40, and 50) for each experimental group with 3 repetitions each. In each density treatment, fruit flies were placed into a glass tube (20 mm diameter, 90 mm high) and one adult spider was introduced, while for a control group, flies were placed in the same glass tubes without introducing spiders.

A water-soaked sponge was placed at the bottom of the tube to maintain humidity. After 24 h, the number of live flies remaining in each glass tube was observed and recorded.

### 2.6. Experiment 4: Effects of Dietary Restriction on Water, Fat and Protein Content in the Body of P. pseudoannulata

We decided to assess individuals three days after maturation to reduce the impact of moult on composition because the mature moult of spiders has an impact on the nutrient composition of individuals. Three days after adults had emerged, 5 males and 5 females from each treatment group were randomly selected. The spiders were weighed and then put in an electric thermostatic drying oven (Shanghai Feiyue Experimental Instrument Co., Ltd., Shanghai, China) for 24 h at 60 °C and then weighed again to determine the water content. The dried spiders were soaked in 5 mL chloroform for 12 h, which was used to dissolve fat. Soaking was repeated three times, and then the spiders were dried for 24 h at 60 °C. The mass was determined immediately, and the fat content was calculated. Then, 5 spiders of a similar size were randomly selected in each of the six treatment groups. The spiders were placed in a centrifuge tube, frozen in liquid nitrogen, and weighed with an electronic balance (BT1251, Sartorius Scientific Instruments, Wuhan, China). The total protein concentration was determined in the homogenate according to the instructions of the total protein assay kit (with standard: BCA method, Nanjing Jiancheng Institute of Bioengineering). Procedures were as follows. The tissue was accurately weighed and added the volume of normal saline was according to the ratio of weight (g): volume (mL) = 1:9. The mixture was mechanically blended to homogenate in an ice-water bath by using a centrifuge at 2500 rpm for 10 min. The supernatant was taken and diluted with normal saline at a ratio of 1:9 to make a 1% tissue homogenate for testing. The working solution is taken in the ratio of reagent I application solution: reagent II = 50:1. The three wells of the microplate were, respectively added with 10 μL of distilled water, 10 μL of standard concentration (524 μg/mL), and 10 μL of test solution, and then each of the three wells was added 250 μL of working solution. The microplate was gently shaken and incubated at 37 °C for 30 min. The absorbance value of each well was recorded with a microplate reader at a wavelength of 562 nm (A_blank_, A_standard_, and A_measured_).

### 2.7. Data Analyses

Data were analysed to satisfy the assumptions for parametric analyses (normal distribution of residuals and homogeneity of error variances) and were expressed as the mean ± standard error (SE). Two-way ANOVA with Tukey’s HSD test was used to analyse the carapace size, body length, body weight, and longevity. One-way ANOVA with Tukey’s HSD test was used to analyse the nutrient content. The Holling II disc equation and the Marquardt method were used to analyse the predation function [36]. Due to the significant error associated with the classical reciprocal method in determining the Na and T_h_ parameters, the Marquardt method was employed for data fitting. And, the R^2^ was used to measure the goodness of fit of the regression model [37,38]. SPSS 21.0, and Origin 9.1 software were used for data analyses, chart making, and graphing.

The data processing formula in experiment 3 was as follows:Na = Ta′N/[(1 + a′T_h_N)].

Note: Na: number of preys being preyed upon; N: density of the prey; T: a total time for one experiment; a′: instantaneous attack rate; T_h_: processing time [39].

The data processing formulas in experiment 4 were as follows:Water content = (wet weight − dry weight)/wet weight × 100%.
Fat content = (dry weight before soaking in chloroform − dry weight after soaking in chloroform)/(dry weight before soaking in chloroform) × 100%

## 3. Results

### 3.1. Experiment 1: Effects of Dietary Restriction on the Growth of P. pseudoannulata

#### Determination of Developmental Period

Dietary restriction levels during the juvenile period could have significant impacts on the total developmental duration of *P. pseudoannulata* (Figure 1). Spiders under diet restriction underwent an additional moult. Groups M and L experience one more instar than group H before appearing the sign of maturity. The total developmental duration of *P. pseudoannulata* in the H group (Male: 89.97 ± 1.96; Female: 94.49 ± 2.24) is lower than that in the M and L groups. Meanwhile, the total developmental duration of *P. pseudoannulata* in the M group (Male: 124.61 ± 4.04; Female: 131.85 ± 4.43) is lower than that in the L group (Male: 144.54 ± 3.96; Female: 150.69 ± 5.14). Furthermore, there was no difference between males and females in the same group in the total developmental duration of *P. pseudoannulata*.

### 3.2. Determination of Carapace Length, Carapace Width, Body Length, and Weight

Dietary restriction at both juvenile and adult had impacts on the size of *P*. *pseudoannulata* individuals (Figure 2). When diet levels in the juvenile period were similar, dietary restriction had limited effects on the width and length of the carapace at the adult period (*p >* 0.05). When diet levels in the adult period were similar, there were significant differences among group H, group M, and group L (*p* < 0.05). The width, length, and weight of the carapace in group H were all higher than those in group M. For group M, the width, length, and weight of carapace were also higher than those in group L. Surprisingly, if feeding conditions are normal during either juvenile or adult, *P*. *pseudoannulata* adults could eventually reach a normal weight.

### 3.3. Experiment 2: Effects of Dietary Restrictions on the Longevity of P. pseudoannulata

Effects of dietary restrictions on the longevity of *P. pseudoannulata* were shown in Figure 3 below. The effects that dietary restrictions had on longevity were more significant during the adult period. To be specific, the longevity of *P. pseudoannulata* in group XH (HH, MH, and LH) was largely lower than that in group XL (HL, ML, and LL), respectively. In other words, when *P. pseudoannulata* suffered dietary restriction during juvenile, they would use more energy that was attained during the adult stage for longevity extension.

### 3.4. Experiment 3: Effects of Dietary Restrictions on the Predation Function of P. pseudoannulata

The predation number of *P*. *pseudoannulata* was fitted to the Holling II disc equation. The Marquardt method was used to estimate the parameters, and the correlation coefficient (r) was used to test the fitting effect.

From the R^2^ of the fitting equation obtained in Table 1, predation of the six experimental groups was significantly positively correlated with the density of fruit flies (*p* < 0.05). The control ability (a′/T_h_) of *P*. *pseudoannulata* under different feeding treatments was HL > MH > ML > HH > LL > LH. The relationship of time (T_h_) to consume prey was LH > LL > HH > ML > MH > HL; the upper limit of predation (1/T_h_) (the number of preys captured by the predator per hour) was: HL > MH > ML > HH > LL > LH. The results indicated that starvation during the juvenile period had a direct impact on the predation function of *P*. *pseudoannulata*, and mild food restriction (M) could increase the predation ability of *P*. *pseudoannulata*. And, there is no significant difference in the instantaneous attack rate of *P*. *pseudoannulata* in the HH, LH, and LL groups. The result of this experiment cannot indicate that the factors above are involved in energy allocation or trade-off.

### 3.5. Experiment 4: Effects of Dietary Restriction on Water, Fat and Protein Content in P. pseudoannulata

The results are shown in Table 2. The total water content in the HL, ML, and LL groups was significantly higher than that of the control group (HH) (*p* < 0.01), and the water content in the MH and LH groups was not significantly different from that of the control group (*p* > 0.05). Compared to the control group (HH), sustained severe dietary restriction (LL) resulted in a significant decrease in the total fat content of *P*. *pseudoannulata* (*p* < 0.05). The body fat content of *P*. *pseudoannulata* in the LH group reached the same level as that of the control group (HH). The total protein concentration in the MH and ML groups was significantly lower than that in the HH group (*p* < 0.01). The total protein concentration in the HL, LH, and LL groups was not significantly different from that of the control group (*p* > 0.05).

## 4. Discussion

Food is important for all animals, without which they cannot survive [40,41]. However, in nature, the phenomenon of dietary restriction is very common, which has a serious impact on the performance of animals. Dietary restriction can also lead to energy allocation. However, few studies have examined the impact of dietary restriction on the development, nutrient content, and predation by *P*. *pseudoannulata*, and the interactions among them are still unclear. The present research on the growth, development, and longevity of *P*. *pseudoannulata* under dietary restriction can help us learn more about the response of *P*. *pseudoannulata* when external food resources are scarce.

Compared with *P*. *pseudoannulata* in the HH group (control group), *P*. *pseudoannulata* in the LL group had a smaller carapace with a longer longevity, while in the LH group, there were characteristics of energy allocation with shorter longevity. These results showed that energy allocation would have different degrees of impact on growth development (carapace length, carapace width, body length, and weight), development duration, and longevity for *P. pseudoannulata*. The predation ability of *P*. *pseudoannulata* was improved under moderate dietary restriction. The protein content of spiders reached the level of the control group (HH group) except the MH and ML groups. The fat content of spiders reached the level of the control group (HH group) except the LL group.

Studies of the missing sector orb weaver *Zygiella x-notata* (Araneidae) and cellar spider *Pholcus phalangioides* (Pholcidae) showed that dietary restriction could prolong the developmental period [42,43]. A similar phenomenon was observed on the effect of dietary restriction in insects. In the study of the mosquito *Anopheles darlingi* in the Amazon basin, juveniles need more energy to complete moulting under dietary restriction, thus prolonging the development time [15]. Our results in experiment 1 also showed that food was the main factor affecting the development period of *P*. *pseudoannulata*, and dietary restriction extended their development period. However, this rule does not always apply to spiders. The wolf spider *Lycosa tarantula*, reared on a restricted diet, reached maturity at a younger age [44].

Many aspects of life are related to body size [8,45]. Food restriction produces intraspecific differences in the adult body size of the solitary bee *Megachile rotundata*, which in turn affects other physical performance characteristics [45]. Body size is also an important factor in evaluating the development of spiders [46,47]. There are species-specific effects of dietary restriction on the size of individual spiders. Carapace size, body length, and weight of *L*. *sclopetaria* after a dietary restriction did not change significantly [48], while the body length and weight of orb weaver spider *Cyrtophora citricola* (Araneidae) after a dietary restriction were significantly reduced compared to the unrestricted group [49].

In our study, the development of *P. pseudoannulata* carapace would be limited if juvenile individuals suffered from dietary restriction. However, as long as *P. pseudoannulata* individuals take in enough energy and nutrition in adulthood to make up for undernutrition during juvenile period, they still could grow normally, and the weight were as same as the normal ones. It indicates that when nutrition is limited during the juvenile period and *P. pseudoannulata* would temporarily develop more slowly until they have sufficient nutritional supplementation in adulthood, which also reflects their energy allocation. However, there were limitations in our study, we did not observe the growth of *P. pseudoannulata* after they took in enough nutrition as a previous study did for wolf spiders, *Pardosa prativaga* [50].

Energy allocation has certain ecological and physiological costs. During a stress period, wolf spiders may reproduce at a subnormal size or increase the developmental period for a whole year; however, if the ecological and physiological costs of energy allocation are lower than either reproducing at a lower size or increasing the developmental period for a whole year, wolf spiders will choose energy allocation [50]. Under dietary restrictions, guppies faced a significant decline in fecundity after energy allocation [20]. This shows that energy allocation is a manifestation of trade-offs. In this study, while continuous dietary restriction (LL group) resulted in significantly prolonged adult life and total longevity, there was a negative impact on adult spider life and overall longevity in the LH group. The results contrasted the observed results of food or calorie restriction of *Larinioides sclopetarius* [48]. This may be due to the dietary impacts on individual longevity, which showed *P. pseudoannulata* would distribute energy they took in during the growth period. The effect of food restriction on longevity extension has been observed in many animals. Dietary restriction shortened the longevity of the blowfly *Chrysomya chloropyga* [51] but prolonged the longevity of adult *Bicyclus anynana* [52]. A better understanding of the trade-off response mechanism of animals suffering dietary restriction can help predict which life history characteristics will recover to a normal level after the food supply increases and which characteristics will show lasting effects.

Our results indicated that the predation function of *P*. *pseudoannulata* was mainly affected by the continuous influence of the dietary restriction level of the juvenile stage. In the juvenile stage, moderate restriction promoted its predation function, indicating that moderate feeding limitation could increase the predatory ability of *P*. *pseudoannulata* on fruit flies. While severe dietary restriction weakened the predation function of *P*. *pseudoannulata*, the predation function was not improved after high feeding was restored in the adult period. Since this experiment was completely conducted in the laboratory, it did not consider natural factors, such as the preference of *P*. *pseudoannulata*, the interference of the prey itself or the influence of the environment [53]. Foraging theory predicts that to compensate for the residual effects of dietary restriction, adult spiders that have experienced a shortage of prey in their juvenile stage must show a greater increase in foraging activities after maturity than those spiders that have not experienced dietary restriction during the juvenile stage. In contrast to expectations, female spiders in the restricted group did not increase their foraging activities [30,54]. Dietary restriction may not be harmful; appropriate dietary restriction may enhance performance such as enhancing tolerance to cold and learning ability [55,56]. However, severe dietary restriction or starvation can have negative effects, possibly because of the difference between short- and long-term stress. Short-term stress is good for improving performance, while long-term stress will weaken performance [57,58]. In this study, the effect of dietary restriction on the predation function of *P*. *pseudoannulata* is more likely to be long-term stress, as predation activities were reduced under severe dietary restriction.

This study found that the total water content of *P*. *pseudoannulata* was affected by the combined effect of dietary restriction levels on the juvenile and adult stages. The total water content in the body could be improved after dietary restriction. Restricting diet during the juvenile stage increases fat storage in adult *Drosophila* [58]. In this study, the fat content of *P*. *pseudoannulata* in the LH group reached the same level as that of the control group (HH group), while the total fat content of *P*. *pseudoannulata* with continuous severe dietary restriction (LL group) decreased, indicating that *P*. *pseudoannulata* also had fat storage after returning to a normal diet. The body fat content of *P*. *pseudoannulata* in the LH group also meets the trend to energy allocation. This study also found that the protein content of *P*. *pseudoannulata* was mainly affected by the food level of the juvenile spider. This effect was continuous and permanent; it would not change due to the adjustment in the food level of the adult spider. The total protein concentrations were decreased in the two moderately restricted groups (MH and ML groups) in the juvenile stage but were not changed in the two severely restricted groups in the juvenile stage (LH and LL groups). This suggests that the effect of dietary restriction on protein concentration depends on the intensity of dietary restriction under certain circumstances. *P*. *pseudoannulata* might have the physiological behaviour of storing protein to prolong longevity under severe dietary restrictions. This is similar to the change in body fat in fruit flies under dietary restriction [59]. The effects of dietary restriction on the nutrients of different animals are species specific. The fat and protein content in adult snow crab *Chionoecetes opilio* increased significantly under dietary restriction, while the water content decreased significantly [60]. The water content of *Babylonia areolata* gradually increased during the 120-day dietary restriction, while the protein and fat content showed a downward trend [61].

In this study, the effects of dietary restriction on the development, longevity, predation, and nutrient content in the body of *P*. *pseudoannulata* were studied. Dietary restriction inhibited the growth rate of *P*. *pseudoannulata*. In MH and LH groups, the longevity of *P*. *pseudoannulata* was significantly shorter than other groups (ML and LL group), which means the individuals would distribute less energy they took in to extend their longevity. The above results reflect that there appears to be a trade-off of *P*. *pseudoannulata* under dietary restriction. In other words, when *P*. *pseudoannulata* individuals suffered dietary restrictions, they would grow more slowly. When the diet resumes in adults, prioritise energy distribution to weight instead of longevity extension. The nutrients in the body of *P*. *pseudoannulata* in the LH group reached the level of the control group (HH group), but there was a significant decrease in its predatory ability. However, the nutrients in *P*. *pseudoannulata* in the MH group decreased and predatory ability increased. However, our results only show the phenomenon of energy allocation, and the effects of this phenomenon on *P*. *pseudoannulata* can be studied further.

## Figures and Tables

**Figure 1 insects-14-00579-f001:**
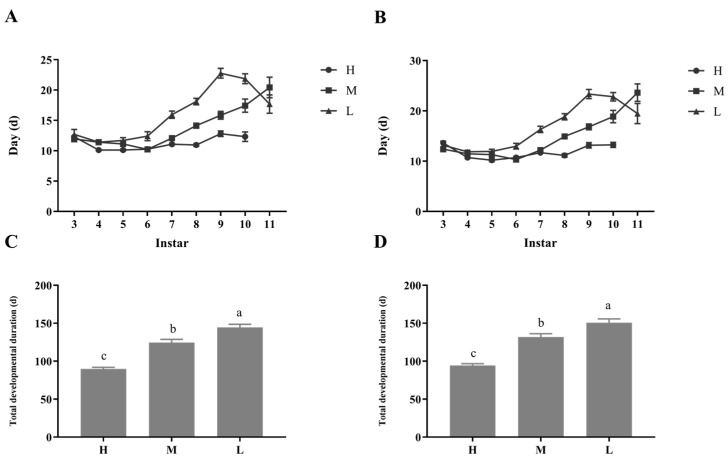
Effect of different dietary restrictions on developmental durations of *P. pseudoannulatas*; (**A**) The days required for male moulting of each instar; (**B**) The days required for female moulting of each instar; (**C**) Total developmental time of male nymphal; (**D**) Total developmental time of female nymphal. H, high feeding; M, medium dietary restriction; L, severely dietary restriction. The differences in lower case letters above each bar indicate significant differences (*p* < 0.05).

**Figure 2 insects-14-00579-f002:**
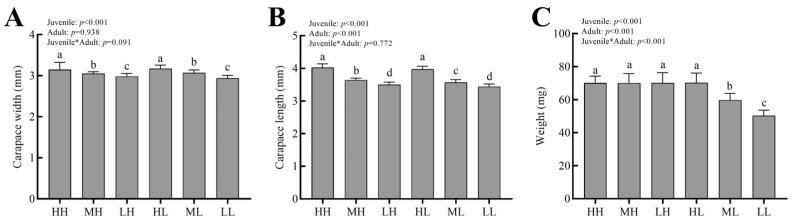
Carapace length, carapace width, and weight of *P. pseudoannulata* under dietary restriction. (**A**) Carapace width; (**B**) carapace length; (**C**) weight. H, high feeding; M, medium dietary restriction; L, severely dietary restriction. HH refers to high feeding at the juvenile and adult stages, HL refers to high feeding at the juvenile stage and severely dietary restriction at the adult stage, etc. Asterisks in (**A**–**C**) indicates that the analysis between the Juvenile and Adult groups. The differences in lower case letters above each bar indicate significant differences (*p* < 0.05).

**Figure 3 insects-14-00579-f003:**
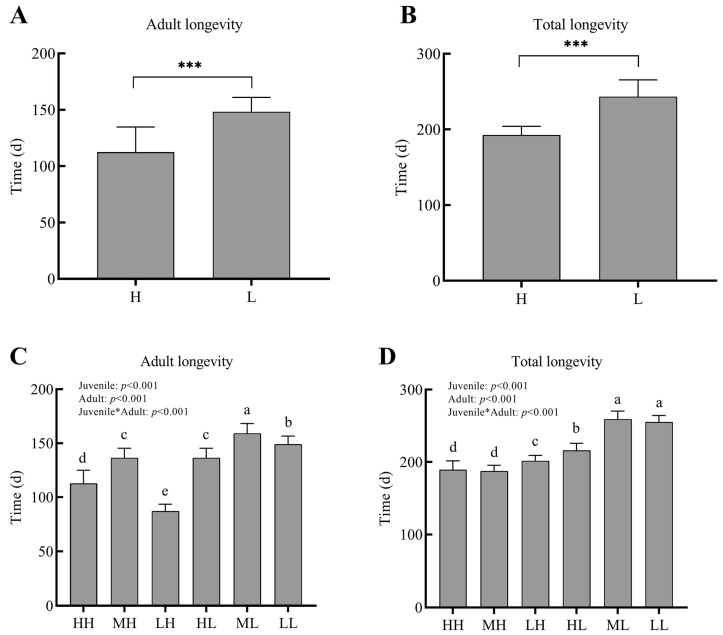
The Adult longevity and Total longevity of *P. pseudoannulata* under different dietary restriction longevity (days). Adult longevity (**A**) and Total longevity (**B**) of *P. pseudoannulata* in H and L group. Adult longevity (**C**) and Total longevity (**D**) of *P. pseudoannulata* in six groups. H, high feeding; M, medium dietary restriction; L, severely dietary restriction. HH refers to high feeding at the juvenile and adult stages, HL refers to high feeding at the juvenile stage and severely dietary restriction at the adult stage, etc. Adult longevity refers to the length of time from maturity to death. Total longevity refers to the time a spider takes from hatching to death. Asterisks in (**A**,**B**) indicate statistically significant differences (*** *p*< 0.001). Asterisks in (**C**,**D**) indicates that the analysis between the Juvenile and Adult groups. The differences in lower case letters above each bar indicate significant differences (*p* < 0.05).

**Table 1 insects-14-00579-t001:** Parameter estimation of the functional response of *P. pseudoannulata* to *D. melanogaster*. H, high feeding; M, medium dietary restriction; L, severely dietary restriction. HH refers to high feeding at juvenile and adult stages, HL refers to high feeding at the juvenile stage and severely dietary restriction at the adult stage, etc. a′: instantaneous attack rate; T_h_: processing time a′/T_h_ is the control ability, 1/T_h_ is the upper limit of predation, and SE is the standard error. Different lowercase letters indicate significant differences (*p* < 0.05).

Treatment	Fitting Equation	R^2^	a′ ± SE	T_h_	1/T_h_	a′/T_h_
HH	Na = 0.89N/(1 + 0.014 N)	0.957	0.89 ± 0.06 ^c^	0.016	62.5	55.63
HL	Na = N/(1 + 0.005 N)	0.986	1 ± 0.03 ^b^	0.005	200	200
MH	Na = 1.2N/(1 + 0.014 N)	0.889	1.2 ± 0.64 ^a^	0.012	83	100
ML	Na = N/(1 + 0.014 N)	0.983	1 ± 0.56 ^b^	0.014	71.43	71.43
LH	Na = 0.95N/(1 + 0.037 N)	0.900	0.95 ± 0.33 ^c^	0.039	25.64	24.36
LL	Na = 0.89N/(1 + 0.016 N)	0.953	0.89 ± 0.14 ^c^	0.018	55.56	49.44

**Table 2 insects-14-00579-t002:** The content of water, fat, and total protein of *P*. *pseudoannulata* under six dietary treatments. H, high feeding; M, medium dietary restriction; L, severe dietary restriction. HH refers to a high feeding at juvenile and adult stages, HL refers to high feeding at the juvenile stage and severely dietary restriction at the adult stage, etc. Different lowercase letters indicate significant differences between dietary restriction treatments (*p* < 0.05).

Treatments	Water (% of Wet Weight)	Fat (% of Dry Weight)	Protein (g/L)
HH	69.92 ± 0.007 ^b^	14.66 ± 0.22 ^a^	4.88 ± 0.88 ^a^
HL	73.85 ± 0.006 ^a^	10.49 ± 0.02 ^ab^	4.36 ± 0.23 ^a^
MH	70.36 ± 0.01 ^b^	10.49 ± 0.02 ^ab^	1.42 ± 0.22 ^b^
ML	75.31 ± 0.01 ^a^	8.5 ± 0.02 ^ab^	1.25 ± 0.13 ^b^
LH	69.94 ± 0.005 ^b^	14.52 ± 0.008 ^a^	3.5 ± 0.23 ^a^
LL	75.19 ± 0.009 ^a^	5.6 ± 0.02 ^b^	3.1 ± 0.15 ^a^

## Data Availability

The data presented in this study are available on request from the corresponding author.

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
