# Peer review of "Energy Allocation of the Wolf Spider *Pardosa pseudoannulata* under Dietary Restriction"

_insects, 2023, doi:10.3390/insects14070579_

Round 1

Reviewer 1 Report

The authors examine the effects diet restriction on size, development time, longevity, predatory ability and nutrient content in the wolf spider Pardosa pseudoannulata. There are a handful of papers on the effects of dietary restriction on spiders, specifically with regard to longevity, and the tie in to predatory behavior is interesting and useful in understanding adaptive responses to food scarcity.  However, I also have some significant critiques of the manuscript. 

Perhaps my biggest problem was “energy input selection.”  This was defined as “Energy input selection is a manifestation of trade-offs that is defined as a physiological process referring to the growth mechanism in which animals return to normal feeding  after a period of dietary restriction or starvation.”  Defining something as a “manifestation” of a “process” “referring” to a “mechanism” is essentially meaningless.  The only concrete part of that definition was returning to normal feeding, and in this experiment, that was determined by the researchers.  The references given at the end of the paragraph do not use this term, and when searching Google, nothing comes up as relevant.  I had no idea what kind of data would support energy input selection and what would refute it.  Because of this, much of the paper did not make sense to me. 

It would also be helpful to better describe what is considered high or low predatory behavior to set up what data would meet those expectations.  This would make the various measures and calculations for the predatory behavior portion of the study more easily digestible. Finally, I would also like to see the analyses redone with a Tukey test for multiple comparisons.  The Duncan test is very permissive compared to other multiple test procedures, and many bars that look equal to me are coming out as statistically different (See Fig. 2B for example). 

Lines 16-17: “…predation of P. pseudoannulata” implies other animals eating P. pseudoannulata, while later in the abstract, it seems that what is meant is predation BY P. pseudoannulata.  Consider changing to “predation by” or “predatory ability.”

Line 20: It is unclear what “energy input selection” means here.  Consider being more explicit. 

Lines 53-57: This definition of “energy input selection” is hard to follow.  And the citations given don’t use the term (also citation 18 is missing a title in the references). 

Lines 58-69: I was unable to understand almost all of this paragraph.  What does “selection” mean here?  Is it a trade-off?  What is the difference between growth and development in this context? What are the developmental costs of anti-predator behavior?  Is the idea that anti-predator behavior trades off with foraging time, resulting in reduce food resources and slower development?  If so, that needs to be made explicit.  What is meant by selection between longevity and body size not being beneficial?  What is the currency of the benefit here?  Are we talking about reproduction? 

Line 71: Spiders in particular are well known to reduce their metabolism under periods of starvation.  It would be worth citing Anderson 1974, Ecology or similar literature. 

Line 71: I don’t understand what “digestibility in the body” means.

Line 82: Does this mean that lacewings with greater dietary restriction took longer to subdue their prey.  Again it would be helpful to be more explicit. 

Line 101: Pardosa should be spelled out here, as it is the first time the species name is used in the main text.  In addition, I’m missing a sentence here that explains that this species, in particular, is common in rice fields and may be an important component to biological control. 

Line 151: By my calculations, you started with 45x5sacs = 225, and 74+75+70 = 219 survived.  That’s a very good survival rate.  It would be helpful to put those numbers in the main text to help point this out to the reader.

Lines 157-158: I don’t understand this sentence.  It seems to say that spiders taken out of the feeding regimen were not taken out of the feeding regimen. 

Line 160: Just stating that the width was taken at the widest point is a sufficient description.

Lines 215-216: I am not familiar with these methods. It would helpful to provide a short description in addition to the reference.

Lines 218-223: It would be helpful to describe the real-world significance of these measures and define what is considered high or low predatory ability. 

Figure 1: The number of graphs here is probably unnecessary and makes the figure hard to read.  The overall pattern seems to be sufficient.  Also why are there no H diet spiders for the 11th instar?  Alternatively, Figure 1 could be a line graph with 3 lines for each diet, with instar on the x-axis and days to the next instar on the y-axis.

Lines 243-252: Does the number of molts to adulthood vary in this species?  Diet can strongly influence the number of molts in other spiders.  In Cyrtophora citricola, even though diet restriction decreased body size, it also increased the number of molts to adulthood, so adults ended up roughly the same size (Yip and Lubin 2016 Biol. J. Lin. Soc.)

Figure 3: Put (days) after “longevity,” rather than “treatment.”

Line 332: It would be better to be cautious here about whether increased predatory behavior is “beneficial.”  That really depends on the fitness consequences for the spider.  It may be that well fed spiders avoid costly and potentially dangerous predatory behaviors when they are well satiated and maintain higher fitness than spiders with increased predatory behavior.

Line 343: This statement seems to be missing a citation.

Lines 353-356: Fig. 2 AB show that this statement is not true.  Adult feeding cannot rescue carapace length and width, which makes sense because carapace shape only changes with additional molts.

372 and 376: The genera names and the type of animal for L. sclopetarius and B. anynana need to mentioned.

Lines 430-432: “In other words, when P. pseudoannulata individuals suffered dietary restriction, they would grow more slowly, and prioritize energy distribution to size development instead of longevity extension.”  What is the evidence for this?  It appears that they live longer and are smaller.   

Discussion: Since the authors frame the system as important for biocontrol, how do the results here influence or potentially influence in the future agricultural practices?

I have noted some parts of the manuscript that were difficult to understand, but I have not commented on proper grammar when the meaning of the sentence was clear.  The English grammar was largely fine, but could use some additional editing.   

Author Response

Dear reviewer:

Thanks for your comment. We carefully revised the manuscript based on these comments. And answered all the questions point by point in the uploaded file.

Reviewer 2 Report

This manuscript provides some interesting results. However, before it can be accepted the comments below must be addressed.

Line 45: Change "roughly" to approximately

Line 52: Please specify the identity of the spiders here, so that the reader does not have to refer to the references to know which taxa were used in previous research.

Line 56: It would be pertinent to reference more lycosid-specific studies on allometry and growth here.

Line 63: Add vertebrates such as before "guppies"

Line 66: It would be nice to reference at least one more study.

Line 70: This sentence is self-evidence and needs to be deleted.

Line 71: "Worse" to "disrupted"

Line 77: Species referring to animals in general or just spiders?

Line 83: Even if rarely reported, this suggests there are at least a few studies, which should be cited.

Line 98: delete "investigations and"

Line 101: Font in incorrect size format

Line 111: For what purposes would the spiders need to be bred in large numbers? For more experiments? Make this clear

Line 116-123: You must include references as to how the spiders were identified, and to ensure they were identified correctly.

Line 130: Write genus out in full.

Line 130: Specify order and family of Tendipes

Line 162: Using an ocular micrometer?

Line 169: "lived from maturity to death", not "taking from mature to death.".

Line 233: "Femal" to "Female"

Line 235: As above

Line 236: As above.

Line 318: So are all the other spiders that occur in rice fields. You need to specify what makes these wolf spiders so special!

Line 335: Add (Araneidae)

Line 335: "garden spider" to "Missing Sector Orb Weaver"

Line 335: Add (Pholcidae)

Line 338: Mosquito to "the Mosquito"

Line 342: Wolf spider to "The Wolf Spider"

Line 343: Study citation needs to be added here

Line 350: The common name of the spider is wrong.

Line 350: Please state that this belongs to the Araneidae.

Line 360: another study to "a previous study"

Line 361: Delete ', that we mentioned before'

Fine, except some instances of "Femal" instead of Female.

Author Response

(The authors gave the same response as above.)

Reviewer 3 Report

Energy input selection of the wolf spider Pardosa pseudoannulata under dietary restriction

Reviewer Comments:

This paper is about the effect of starvation and low food supply on the development and predatory behavior of wolf spiders. They do this by raising wolf spiderlings from the egg sac and then feeding them at 3 levels, then as adults feeding them at 2 levels. This is done to see how lack of food affects their size, fat content, and predatory behavior. Raising spiderlings is extremely difficult and I am impressed by their ability to keep so many alive, through their trials.

               This paper successfully looks at the differences in predatory behavior and hunting behavior of organisms who grew up with food shortages. These data confirm many of the previous observations of weight loss and fat content from mammals and insects. Interestingly enough, they found that lower access to food when young lowered the amount of hunting the spider did in adult hood. I think this paper is important as it provides us with insight into how these starving conditions affect an active hunting species such as wolf spiders.

Major Comments:

I am not a physiologist so some of the terminology was a little confusing to me. Some of the terms are vague and could be interpreted in multiple ways (especially when taking data), such as processing time. If important concepts such as that could be better described that would help with clarity. If anything was defined and I missed it, please ignore my comment.

I think the discussion could be more succinct in their comparison of their results to previously found material. I think their work is more important than the information from previous studies.

Equations in the methods could be cleaned up, Lines 219-227. I think these equations could be made easier to read for the audience.

The Figures are useful, but I think it would help if you added an overall summary sentence or two to the figure legends; to let the reader know what trends to notice. There are many graphs.

I think the English is totally readable and understandable, but some phrases could be fixed to make the methodology clearer (some examples below)

Minor Comments:

General Question: How would it change if the spiders were allowed to mate? In a natural setting large females would then lay eggs and protect their egg sacs. This would affect their overall size and fat content and perhaps limit their predatory behavior? Is that a factor that might be interesting (maybe in the future)?

Line 36: Descriptive statements? Terms?

Line 126: Does this mean 45 spiderlings from each egg sac or 45 total? Perhaps change to “randomly selected from each egg sac and…”

Line 143: Perhaps better to say, “was 30 or greater.” Or “at least 30.” Instead of not less than 30.

Line 168: How do you measure Longevity daily? Wording

Also, what is developmental period, can we have a definition? Or did I miss it?

Line 169-170: To clean up the language a little from “Adult longevity refers to the time a spider taking from mature to death” change to “Adult longevity refers to the length of time from maturity to death.”

Also change next sentence to “Total longevity refers to the length of time from hatching to death.”

Line 219-227: Note for editor; I think these equations could be cleaned up and made easier to read for the audience.

Line 231: Sorry if I missed it, but is developmental duration defined in the introduction?  I assume its time from childhood to adulthood?

Figure 1: Even when I zoom in its hard to read the small text of the stats. Are they important to the story? If so, can you make the important ones bigger? Or if not, can we just move them to a supplementary document.

I think it would help if you added an overall summary sentence or two to the figure legend to let the reader know what trends to notice. There are many graphs here.

Figure 2: stats easier to see here. Maybe a summation sentence in the figure legend.

Figure 3: Line 271-273: Same change as line 169. Maybe a summation sentence in the figure legend.

Line 283: What is the upper limit of predation? What’s that definition? (I thought it was how much they ate but some spiders had 200 but I thought the highest density of flies was 50. So, I am a little confused.

Line 384: Is it possible this is tied to changes in their overall metabolism? They have lowered their metabolism and kept it low into adult hood?

Line 435-437: I don’t think you need the justification “Our results will provide an empirical…” I think your work on development and nutrient is strong enough to stand on its own.

Line 126: Does this mean 45 spiderlings from each egg sac or 45 total? Perhaps change to “randomly selected from each egg sac and…”

Line 143: Perhaps better to say, “was 30 or greater.” Or “at least 30.” Instead of not less than 30.

Line 168: How do you measure Longevity daily? Wording

Also, what is developmental period, can we have a definition? Or did I miss it?

Line 169-170: To clean up the language a little from “Adult longevity refers to the time a spider taking from mature to death” change to “Adult longevity refers to the length of time from maturity to death.”

Also change next sentence to “Total longevity refers to the length of time from hatching to death.”

Author Response

(The authors gave the same response as above.)

Round 2

Reviewer 1 Report

The paper still needs moderate editing for English conventions. 

Author Response

Dear reviewer:

Thanks for your again comment. We also carefully revised the manuscript based on these comments. And answered all the questions point by point in the uploaded file.
